# Identification and Characterization of *Janthinobacterium svalbardensis* F19, a Novel Low-C/N-Tolerant Denitrifying Bacterium

**Yinyan Chen [1], Peng Jin [2],\* , Zhiwen Cui [3], Tao Xu [3], Ruojin Zhao [1] and Zhanwang Zheng [1,3,\*]**

1   School of Environmental & Resource, Zhejiang A & F University, Hangzhou 311300, China; cyy0626@foxmail.com (Y.C.); RuojinZhao@hotmail.com (R.Z.)
2   The College of Agricultural and Food Sciences, Zhejiang A & F University, Hangzhou 311300, China
3   Zhejiang Shuangliang Sunda Environment co., LTD, Hangzhou 310000, China; cuizhiwen@foxmail.com (Z.C.); xutao@foxmail.com (T.X.)
\*   Correspondence: jinpeng@zafu.edu.cn (P.J.); zhengzw@zafu.edu.cn (Z.Z.); Tel.: +86-571-18368187965 (P.J.); Fax: +86-571-88218710 (P.J.)

**Abstract:** Herein, we isolated *Janthinobacterium svalbardensis* F19 from sludge sediment. Strain F19 can simultaneously execute heterotrophic nitrification and aerobic denitrification under aerobic conditions. The organism exhibited efficient nitrogen removal at a C/N ratio of 2:1, with an average removal rate of 0.88 mg/L/h, without nitrite accumulation. At a C/N ratio of 2, an initial pH of 10.0, a culturing temperature of 25 °C, and sodium acetate as the carbon source, the removal efficiencies of ammonium, nitrate, nitrite, and hydroxylamine were 96.44%, 92.32%, 97.46%, and 96.69%, respectively. The maximum removal rates for domestic wastewater treatment for ammonia and total nitrogen were 98.22% and 92.49%, respectively. Gene-specific PCR amplification further confirmed the presence of *nap*A, *hao*, and *nirS* genes, which may contribute to the heterotrophic nitrification and aerobic denitrification capacity of strain F19. These results indicate that this bacterium has potential for efficient nitrogen removal at low C/N ratios from domestic wastewater.

**Keywords:** domestic wastewater; low C/N ratio; nitrogen removal; pathway genes; simultaneous nitrification–denitrification

## 1. Introduction

Excessive nitrogen discharge from aquaculture, agricultural fertilizers, livestock waste, domestic wastewater, and industrial effluents leads to eutrophication, affecting the balance of natural water ecosystems and impacting human health [1–3]. Ammonium, nitrate, and nitrite are the principal pollutants in more than 40.0% of surface water, resulting in substandard drinking water in China and elsewhere [4–6]. Therefore, lowering nitrogen levels in wastewater is important.

Biological processes, the most efficient and cost-effective method among nitrogen removal methods [7,8], are the most widely used for wastewater [9,10]. Traditional microbial domestic wastewater treatment is dependent on aerobic autotrophic nitrifying bacteria that convert ammonia to nitrate ($NH_4^+ \rightarrow NH_2OH \rightarrow NO_2^- \rightarrow NO_3^-$), followed by anaerobic denitrifying bacteria that convert nitrate to nitrogen gas ($NO_3^- \rightarrow NO_2^- \rightarrow NO \rightarrow N_2O \rightarrow N_2$) [7,11]. The main disadvantages of traditional biological nitrogen removal are the complexity of the process, high energy consumption, and the need to add carbon sources. By comparison, simultaneous nitrification and denitrification (SND) processes are effective treatments with low energy consumption [12–14]. In SND, nitrification and denitrification occur concurrently in the same reaction chamber due to the actions of bacteria capable of heterotrophic nitrification and aerobic denitrification [15], which are continually being discovered [16,17]. However,

the efficient treatment of wastewater with a low C/N ratio (<4:1) is a barrier, making it difficult to meet the treatment requirements for nitrogen removal from domestic sewage under low nutrient conditions such as those found in typical domestic wastewater in China [18,19]. Therefore, it is necessary to develop novel microbial resources with excellent tolerance of extreme environments to remove nitrogen efficiently under low-C/N conditions, thereby achieving efficient simultaneous nitrification and denitrification supporting nitrogen removal [20,21].

In this study, a novel low-C/N- and pH-tolerant efficient aerobic denitrifying bacterium, *Janthinobacterium svalbardensis* F19, was isolated and characterized. The nitrogen removal performance of the strain was investigated under aerobic conditions, and the influence of various factors was tested. Finally, the application potential of this strain in a low-C/N domestic culture system was explored. This strain exhibited excellent capacity for simultaneous heterotrophic nitrification and aerobic denitrification, making it potentially suitable for nitrogen removal under low-C/N conditions.

## 2. Material and Methods

### 2.1. Media

The enrichment medium (EM) consisted of (g/L): 1.00 g yeast extract, 5.00 g peptone, 2.00 g $NH_4Cl$, and 2.00 g $KNO_3$. The nitrification medium (NM) consisted of (g/L): 2.50 g glucose, 0.38 g $NH_4Cl$, 0.50 g $KH_2PO_4$, 0.50 g $K_2HPO_4$, 0.20 g $MgSO_4 \cdot 7H_2O$, and 2.00 mL trace elements. Denitrification medium 1 (DM1) consisted of (g/L): 2.50 g glucose, 0.72 g $KNO_3$, 0.50 g $KH_2PO_4$, 0.50 g $K_2HPO_4$, 0.20 g $MgSO_4 \cdot 7H_2O$, and 2.00 mL trace elements. DM2 consisted of (g/L): 2.50 g glucose, 0.05 g $NaNO_2$, 0.50 g $KH_2PO_4$, 0.50 g $K_2HPO_4$, 0.20 g $MgSO_4 \cdot 7H_2O$, and 2.00 mL trace elements. DM3 consisted of (g/L): 2.50 g glucose, 0.21 g hydroxylamine hydrochloride, 0.50 g $KH_2PO_4$, 0.50 g $K_2HPO_4$, 0.20 g $MgSO_4 \cdot 7H_2O$, and 2.00 mL trace elements. The trace element solution contained (g/L): 50.00 g EDTA-2Na, 2.20 g $ZnSO_4$, 5.50 g $CaCl_2$, 5.06 g $MnCl_2 \cdot 4H_2O$, 5.00 g $FeSO_4 \cdot 7H_2O$, 1.10 g $(NH_4)_6MO_7O_{24} \cdot 4H_2O$, 1.60 g $CuSO_4 \cdot 5H_2O$, and 1.60 g $CoCl_2 \cdot 6H_2O$ at pH 7.0. Ammonium chloride, ammonium nitrate, sodium nitrate, and sodium nitrite were added as the nitrogen source as indicated. The initial pH of all the media mentioned above was adjusted to 7.0.

### 2.2. Isolation of Strains

The sludge sediment was sampled from the sewage treatment plant of Lin'an, Hangzhou, China. The medium was adjusted to pH 7.0 and autoclaved for 20 min at 115 °C. Ten milliliters of sludge were added to 90 mL of sterile EM in a 500-mL flask and then shaken at 30 °C and 180 rpm for 3 days to enrich for heterotrophic bacteria accumulation. Subsequently, 10 mL of culture was inoculated into another sterile flask containing fresh NM, respectively, and incubated using the same operation. This process of subculture was repeated three times. The enrichment cultures were diluted via five serial 10-fold dilutions with sterilized water and then distributed into agarose plates containing NM, respectively. After incubation at 30 °C for 2 days, colonies from the media were identified. Then, the isolates were inoculated into NM and incubated at 30 °C with constant shaking at 200 rpm. The nitrogen removal performances of aerobic denitrification and heterotrophic nitrification of the isolated strains were finally evaluated in NM. The strain with excellent nitrification and denitrification ability was obtained and named F19 for further study.

### 2.3. Bacterial Identification and Denitrification Gene Amplification

The 16S rRNA gene of the isolated F19 was amplified using bacterial universal primers F27 (AGAGTTTGATCMTGGCTCAG) and R1492 (TTGGYTACCTTGTTACGACT). PCR was carried out as follows: 5 min at 94 °C, 30 cycles of 1 min at 94 °C, 1 min at 53 °C, 1.5 min at 72 °C, and a final step of 10 min at 72 °C. Primers *hao*F1 (TGCGTGGARTGYCAC) and reverse primer *hao*R3 (AGRTARGAKYSGGCAAA) were used for the amplification of the HAO gene fragment, conducted as previously described [22]. Primers NAP1 (TCTGGACCATGGGCTTCAACCA) and NAP2 (ACGACGACCGGCCAGCGCAG)

were used for *napA* amplification [23]. Primers JBNirS-F2 (CGTGGTGGGAAAYTAYTGGCCKCC) and JBNirS-R1 (CAYGAYGGHGGHTGGGAC) were used for *nirS* amplification. PCR products were run and visualized on a 1% agarose gel and by ethidium bromide staining. The PCR products were purified and sequenced by Shanghai Sangon Biological Engineering Technology & Services Co. Ltd., China. Finally, sequences were compared to other relevant microorganisms in GenBank by BLAST and submitted to the databases.

## 2.4. Assessment of Nitrification and Denitrification Performances

F19 cells were precultured in NM, then incubated at 30 °C with shaking at 200 rpm for 16 h, and next used as the seed cultures. To evaluate the simultaneous nitrification and denitrification capabilities, 100 mL of the modified medium with ammonium chloride as the sole nitrogen source was inoculated with 1 mL of F19 seed culture and then cultured at 20 °C with shaking at 200 rpm.

The influencing factors, including C/N ratios, pH, temperature, and carbon sources, on nitrogen removal from F19 were analyzed with separate experiments. In C/N ratio experiments, the C/N ratios were set as 1, 2, 5, 6, and 10, respectively. To determine the optimal pH, nitrogen removal from F19 was tested in media ranging in pH from 5.0 to 11.0. For the temperature experiments, the culture temperatures were 5, 10, 15, 20, 25, 30, and 35 °C. Five carbon sources, including sodium acetate, glucose, sodium bicarbonate, trisodium citrate, and ethanol, were used to investigate the nitrogen removal effects.

For nitrification analyses, 500-mL flasks containing 100 mL of the modified medium, with an initial concentration of 40 mg $NH_4^+$-N/L used as the sole nitrogen source, were inoculated with 1 mL of F19 seed culture and cultured at 20 °C with shaking at 200 rpm. For denitrification analyses of strain F19, potassium nitrate (40 mg/L), sodium nitrite (10 mg/L), and hydroxylamine (40 mg/L) were used as the sole nitrogen sources, respectively.

For the effects of concentration of ammonium, the capability of high-strength ammonium removal by strain F19 was evaluated using the initial ammonium concentrations of 30, 50, 100, 400, 800, 1000, and 2000 mg/L as the sole nitrogen source.

All of the above experiments were conducted with inoculation of 1 mL of F19 seed culture. Unless otherwise stated, all the experiments were conducted at an initial ammonium concentration of 40 mg/L, C/N of 2, initial pH of 7.0, culturing temperature of 20 °C, and with the carbon source as glucose. The optical density at 600 nm ($OD_{600}$) was used to monitor cell growth. The concentration of ammonium, nitrite, and nitrate were measured periodically as previously described [24]. All the assays were performed in triplicate.

## 2.5. Assessment of Nitrogen Removal in Domestic Wastewater

To test the potential application of the strain F19 in low-C/N domestic wastewater, an integrated bioreactor (500 L) for sewage treatment was designed and installed outdoors. The strain F19 was cultured in 20 L of NM for 16 h as seed culture. Then, 200 L of domestic wastewater from the living quarters of Zhejiang A & F University was added in the culture system, and next, 20 L of F19 seed culture was added. The culture chamber was aerated with an air flow rate of 2.0 vvm for 24 h. Subsequently, domestic wastewater incessantly flowed into the chamber at a constant flow rate of 0.28 L/min and up to 400 L of total volume. According to the design, the treated water in the chamber at more than 400 L will automatically overflow out with the same rate, which effectively ensures that all inflow water was treated by strain F19 for 24 h. Air flow rate in the aerobic zone was regulated consistently with the dissolved oxygen (DO) levels at 4.5 ± 0.2 mg/L, and in the anaerobic zone with the DO levels at 0.5 ± 0.1 mg/L. The experiment was carried out for one month, from May 4 to June 4, 2018. After the 20th day, glucose was added with a nitrogen and carbon ratio of 1:2. Nitrogen removal efficiency of the bacterium was measured (ammonium, nitrite, nitrate, total nitrogen (TN), and chemical oxygen demand (COD)) at regular intervals of 2 days for a period of 30 days. During the experimental period, neither water change was made nor any external carbon source added.

### 2.6. Analytical Methods

The cell density was measured at $OD_{600}$ nm using an ultraviolet spectrophotometer. Ammonium, nitrate, nitrite, and hydroxylamine were detected using the supernatant after samples were centrifuged at 3000 rpm for 3 min. Ammonium concentration was calculated by the absorbance value at 420 nm. Briefly, the nitrate was determined following a spectrophotometric method to differentiate the difference between $OD_{220}$ and $2 \times OD_{275}$. Nitrite nitrogen was determined at wavelengths of 540 nm after adding 1 mL of chromogenic reagent, including (per L) 100 mL phosphoric acid, 40.00 g sulfanilamide, and 2.00 g N-(1-Naphthyl) ethylenediamine dihydrochloride. Hydroxylamine was determined at wavelengths of 520 nm after adding 1.3 mL 6% sodium acetate, 0.2 mL 1% p-aminobenzenesulfonic acid, 0.1 mL 1.3% iodine, 50 μL 0.4 mol/L sodium thiosulfate, and 40 μL 0.6% alpha-Naphthylamine. DO was determined by a dissolved oxygen meter (Bante 820, Bante instruments, China).

### 2.7. Statistical Analysis

The ammonia, nitrate, and nitrite removal ratio formula was $(\rho_0 - \rho_n)/\rho_0 \times 100\%$, where $\rho_0$ is the initial nitrogen concentration and $\rho_n$ is the final concentration of $NH_4^+$-N, $NO_2^-$-N, $NO_3^-$-N, and $NH_2OH$ at $n$ hour. Statistical analysis and graphical work were carried out by using Excel and Origin 9.0. To determine the evolutionary relationship of this strain F19 with established denitrifying bacterium, the 16S rRNA sequence of F19 was compared to 18 representative bacterial denitrifies from the nonredundant (NR) nucleic acid sequence database (NCBI), using neighbor-joining phylogenetic analysis. One thousand bootstrap replications were performed using the MEGA software (MEGA 5.0).

### 2.8. Nucleotide Sequence Accession Numbers

The 16S rDNA nucleotide sequences have been deposited in the GenBank database under accession number MH628667 of strain F19.

## 3. Results and Discussion

### 3.1. Isolation and Identification of F19

In our preliminary screening work, nine strains were isolated, and their nitrogen removal capabilities under aerobic conditions were examined. A strain exhibiting more than 90% ammonium and TN removal efficiency within 24 h without nitrite accumulation was identified and named F19. Strain F19 also exhibited the highest ammonium removal efficiency under a low C/N ratio (<4:1). A fragment of 16S rDNA ~1400 bp in length was obtained by PCR, sequenced, and submitted to the GenBank database under accession number MH628667. BLAST homology searches against GenBank indicated that the novel strain F19 was most closely related to *J. svalbardensis* (100% similarity). To determine the genus relationship, a neighbor-joining phylogenetic tree was subsequently constructed based on the 16S rRNA gene sequence of the isolate and those of phylogenetically related strains and other strains with nitrogen removal capability (Figure 1). The results suggested that strain F19 was a member of the genus *Janthinobacterium*.

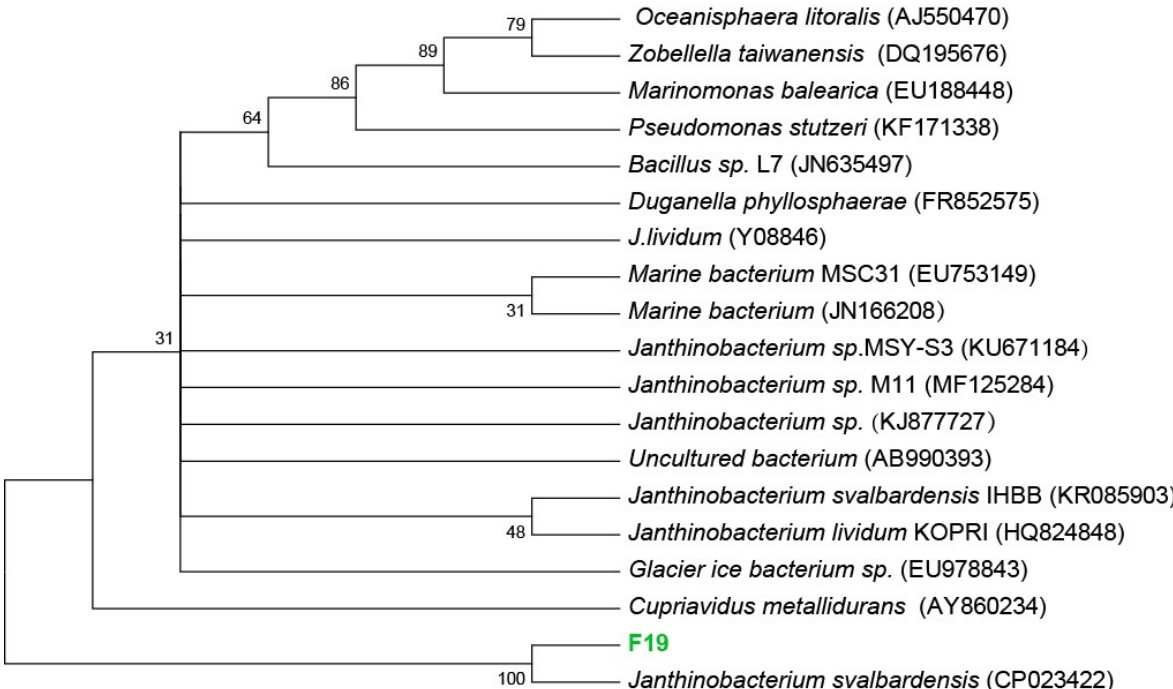

**Figure 1.** Phylogenetic tree of strain F19 and close relatives based on 16S rDNA gene sequences.

*3.2. Factors Affecting Nitrogen Removal by Strain F19*

3.2.1. Effects of C/N Ratio

Figure 2A shows the effects of C/N ratio on the nitrogen removal efficiency. The utilization of ammonium was analyzed with various C/N ratios from 1 to 10 in batch experiments over 48 h. As the C/N ratio declined from 10 to 2, $NH_4^+$-N removal dramatically declined from 97.21% to 78.12% after 36 h of cultivation. When the C/N ratio was further decreased to 1, the $NH_4^+$ removal efficiency was only 35.97% after 48 h of incubation. However, more than 78.12% nitrogen removal activity remained at a C/N ratio of 2 (Figure 2A). The $NH_4^+$ removal efficiency of strain F19 exhibited a lower C/N ratio tolerance than has been observed for some other heterotrophic denitrifiers [25–27]. Furthermore, the accumulation of nitrite was analyzed, and it was not detected with increasing C/N ratio (data not shown). The differences in removal efficiency of $NH_4^+$-N could be due to the consequent delay in cell growth at low C/N ratios (Figure 2B). These results suggested that strain F19 was an excellent heterotrophic nitrification–aerobic denitrification denitrifier, similar to *Agrobacterium sp.* LAD9 [28] and *Alcaligenes faecalis* [29]. Therefore, this strain may be suitable for nitrogen removal from domestic wastewater with a low C/N ratio.

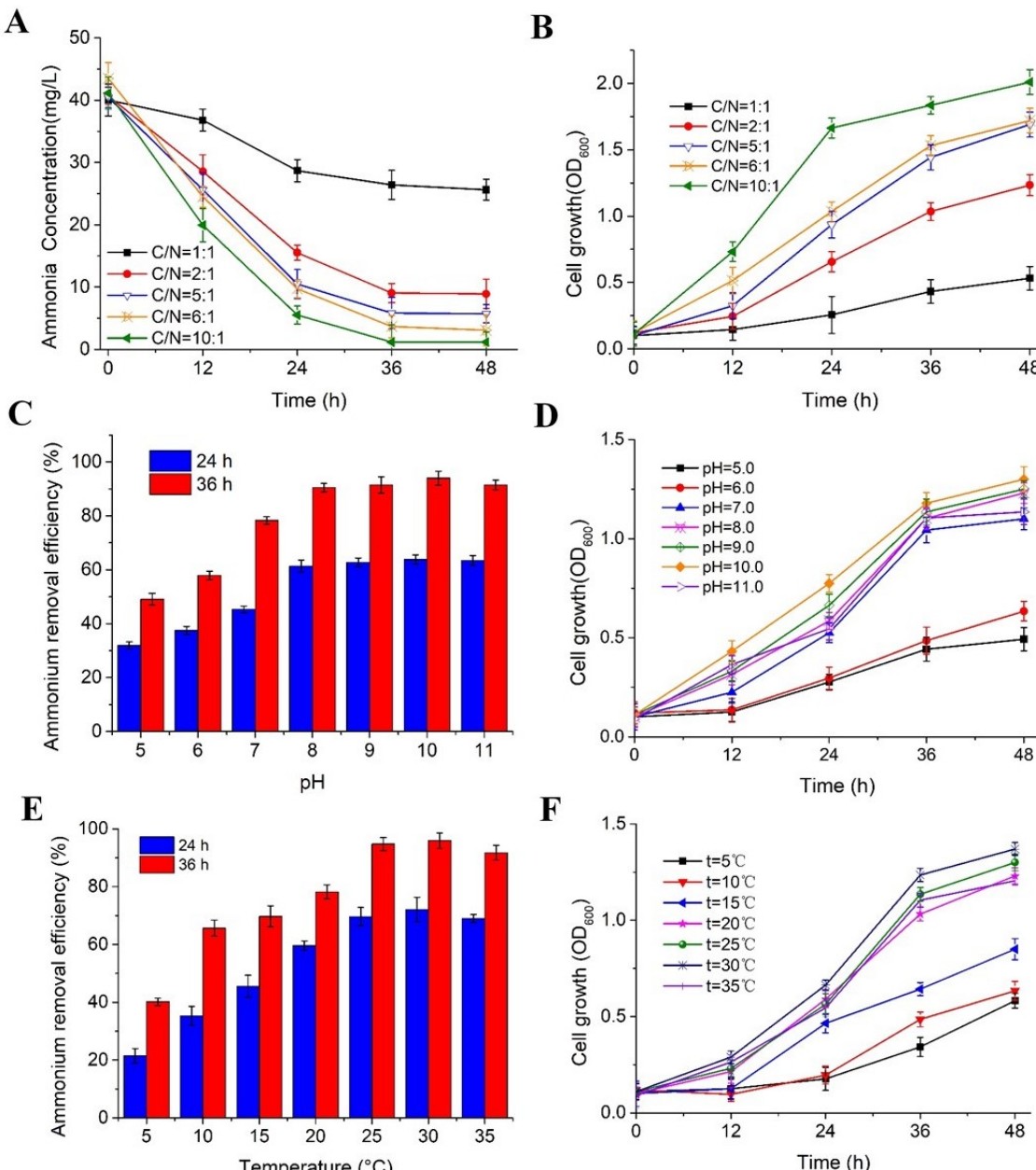

**Figure 2.** Analysis of factors affecting the nitrogen removal capacity and cell growth of strain F19. (**A**,**B**) C/N ratio. (**C**,**D**) pH. (**E**,**F**) Temperature and time.

### 3.2.2. Effects of pH and Temperature

The effect of pH on the nitrogen removal efficiency capacity of F19 was investigated between pH 5.0 and 11.0. F19 was found to be an alkaline-tolerant bacterium with a maximal $NH_4^+$-N removal efficiency of 93.99% and maximal cell growth at pH 10.0 (Figure 2C). The strain was remarkably stable over a wide pH range. After incubating at pH values ranging from 5.0 to 11.0 for 36 h, strain F19 retained residual nitrogen removal activity of more than 50% and 90% at 5.0–7.0 and 8.0–11.0, respectively, compared with controls. However, acidic conditions were tremendously harmful to cell growth; at pH 5, the cell density was only 25% that of the maximal absorbance at 600 nm ($OD_{600}$ value), resulting in a residual nitrogen removal rate of only 50.07%.

This result suggested that strain F19 had strong alkaline resistance, which was beneficial for cell growth under alkaline conditions. The efficient removal of ammonium under alkaline conditions

may be due to the presence of greater quantities of free $NH_3$ in alkaline medium, which is especially conducive to nitrification [30]. This result is consistent with some other heterotrophic bacteria [25] and suggests that the pH tolerance of nitrogen removal by F19 may be superior to that of most previously characterized aerobic denitrifiers [19,31]. This indicates that strain F19 may have potential for use in alkaline wastewater treatment, especially for the efficient bioremediation of alkaline soils [32,33]

Figure 2E shows that the optimal temperature was 30 °C, with a maximal $NH_4^+$-N removal rate of 95.91% after 36 h, and a similar removal rate was measured at temperatures between 25 and 35 °C. At lower temperatures, strain F19 exhibited relatively high removal efficiency; even at 10–20 °C, the enzyme maintained almost 60% of its maximal activity. When the temperature was lower than 5 °C, the removal rate with 40 mg/L $NH_4^+$-N was almost 40% of maximal activity, with a $NH_4^+$-N removal rate of 0.44 mg/L/h. Notably, the maximum $OD_{600}$ value at 5 °C was 0.58 after 48 h, with a maximum specific growth rate of 0.01–0.02 $OD_{600}$/h at a C/N ratio of 2:1. This growth rate was much higher than the rate of 0.0039 $OD_{600}$/h achieved by *Janthinobacterium sp.* M-11 at 2 °C [27] and will have greater potential application. The cell growth rate is of great relevance to cell colonization during nitrogen metabolism [34]. The number of species employed in these previous reports for water treatment is not suitable for the actual conditions needed for domestic sewage treatment [35]. By contrast, the relatively fast growth rate of strain F19 at low temperatures indicates potential for use in existing mainstream wastewater treatment processes, making it favorable for most areas of China during winter and spring.

### 3.2.3. Effects of Carbon Sources

The effects of different carbon sources on cell growth and nitrogen removal efficiency were assessed. As shown in Figure 3, cell growth of strain F19 was obviously restrained with ethanol or trisodium citrate as the sole carbon source (Figure 3A), and the corresponding removal efficiencies were significantly decreased to 39.90% and 22.27%, respectively (Figure 3B). When bicarbonate was used as a carbon source, the nitrogen removal rate was only 48.03%. With sodium acetate or glucose as the sole carbon source, the concentration of ammonium decreased from 43.35 to 2.25 mg/L, and from 44.33 to 9.35 mg/L, respectively, with corresponding nitrogen removal efficiencies of 94.81% and 78.91%, respectively (Figure 3B). The highest ammonia nitrogen removal rate was obtained with sodium acetate as the carbon source, and this was used in subsequent experiments.

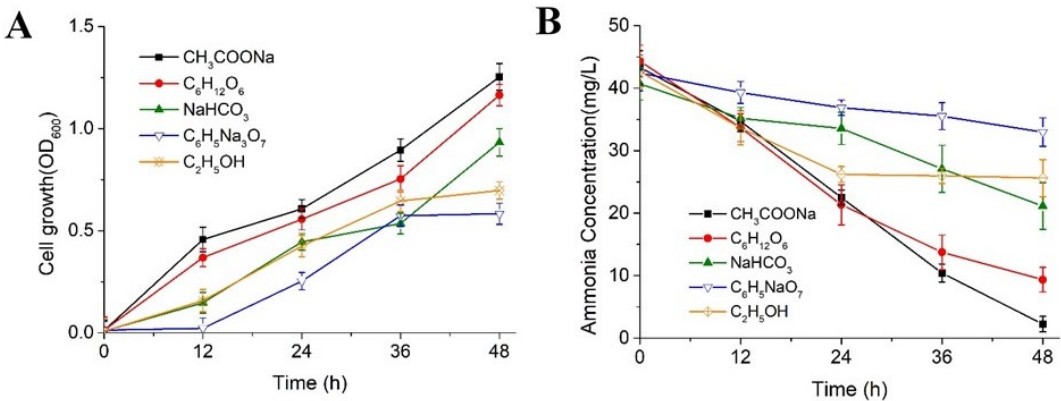

**Figure 3.** Effects of different carbon sources on the nitrogen removal capacity of strain F19. (**A**) Cell growth. (**B**) Ammonia removal.

### 3.3. Analysis of Nitrification and Denitrification Capacity

To characterize the heterotrophic nitrification and denitrification capacities of strain F19, ammonium nitrate, sodium nitrite, and hydroxylamine hydrochloride were used as nitrogen sources at a C/N ratio of 2, an initial pH of 10.0, a culturing temperature of 25 °C, and with sodium acetate as the carbon source. The cell growth and ammonium nitrogen removal performance are shown in

Figure 4. A lag period was observed initially, and the $OD_{600}$ value of strain F19 increased slowly from 0.01 to 0.52 after 24 h. Cell growth then entered the logarithmic phase and increased significantly from 0.52 to 1.26 at 48 h. $NH_4^+$-N at an initial concentration of 46.41 mg/L $NH_4^+$-N was rapidly reduced to 1.65 mg/L, and the corresponding removal efficiency reached 96.44% after 72 h of cultivation. The initial 46.35 mg/L nitrate was decreased to 3.56 mg/L after 36 h, the removal efficiency reached 92.32%, and the removal rate was 1.19 mg/L/h (Figure 4A). Moreover, accumulation of the nitrite intermediate was not detected. These experimental results indicate that strain F19 could conduct heterotrophic nitrification and aerobic denitrification simultaneously under aerobic conditions and efficiently carry out ammonia removal without nitrite accumulation during nitrogen conversion.

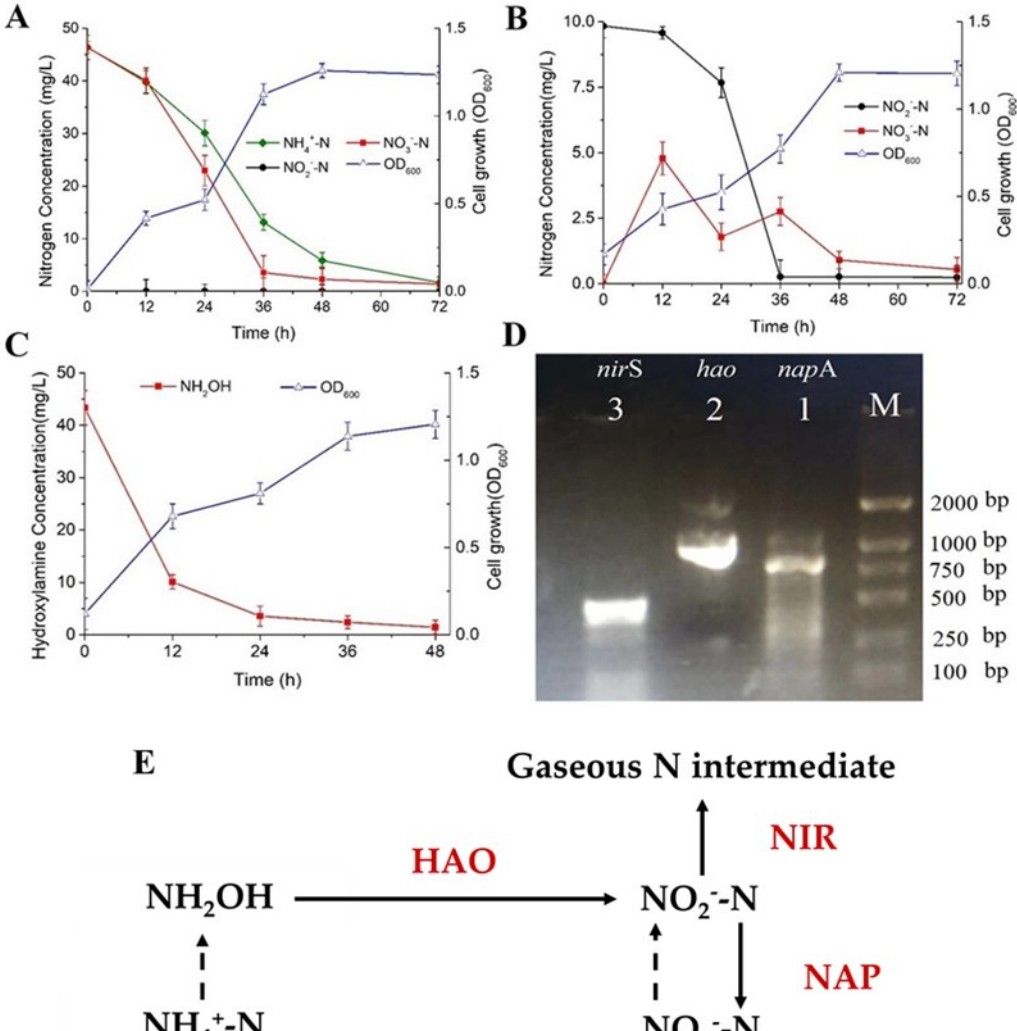

**Figure 4.** Cell growth and nitrogen removal characterization of strain F19, and demonstration of simultaneous heterotrophic nitrification and aerobic denitrification. (**A**) Nitrification ability of F19 in ammonium nitrate medium. (**B**) Denitrification ability of F19 in nitrite and (**C**) hydroxylamine medium. (**D**) PCR amplification of periplasmic nitrate reductase (*napA*), hydroxylamine oxidoreductase (*hao*), and nitrite reductase (*nirS*) genes. (**E**) Predicted nitrogen utilization pathway in *Janthinobacterium svalbardensis* F19.

To investigate the denitrification capacity of strain F19, $NO_2^-$-N at an initial concentration of 9.84 mg/L and $NH_2OH$ at an initial concentration of 43.35 mg/L were used as nitrogen sources in DM2 and DM3, respectively. Using nitrite as a nitrogen source, excellent nitrite nitrogen removal efficiency was observed for strain F19. $NO_2^-$-N was decreased from 9.84 to 0.25 mg/L within 36 h

of incubation (Figure 4B). Approximately 97.46% of nitrite was removed, and the corresponding maximum removal rate was 0.27 mg/L/h, higher than observed previously for *Pseudomonas stutzeri* and *Paracoccus denitrificans* [36]. Unexpectedly, significant nitrate accumulation was observed during the denitrification of nitrite, but this was removed gradually with decreasing nitrite. Previous reports have indicated that large amounts of nitrite nitrogen can inhibit bacterial growth and thereby repress their nitrification and denitrification activities [19]. However, our results showed that the bacterial growth and denitrification capacity of strain F19 were not affected by the concentration of nitrite. This could indicate the countercurrent production of nitrate in the nitrite denitrification process via a unique pathway for nitrite detoxification. When hydroxylamine was used as the sole nitrogen source, strain F19 achieved a similarly high removal efficiency of 96.69%, and $NH_2OH$ was decreased from the initial 43.35 to 1.43 mg/L within 24 h (Figure 4C). Taken together, these results suggest that strain F19 possesses broad substrate utilization capabilities and efficient aerobic denitrification capacity. To further investigate the possible pathway of simultaneous nitrification and aerobic denitrification, and the enzymes involved, hydroxylamine oxidoreductase (HAO), periplasmic nitrate reductase (NAP), and nitrite reductase (NirS), were used as functional biomarkers that are related to the physiology of nitrifying and denitrifying bacteria [37]. PCR amplification of these nitrifying and denitrifying enzyme-encoding genes resulted in 992,877 and 500 bp products for *nap*A, *hao*, and *nir*S, respectively (Figure 4D). These results confirmed the presence of HAO, NirS, and the cytochrome cd1-containing nitrite reductase, all of which may contribute to the heterotrophic nitrification and aerobic denitrification capacity of strain F19. The HAO gene encoded hydroxylamine oxidase, which oxidized hydroxylamine to nitrite, respectively. The *nap*A gene encoding periplasmic nitrate reductase, which reduces nitrate to nitrite under aerobic conditions, was amplified to confirm the aerobic denitrification. The NirS gene encoded cytochrome cd1-containing nitrite reductase *nir*S, which reduces nitrite to nitric oxide (Figure 4E).

### 3.4. Tolerance to High-Strength Ammonium

The capacity for high-strength ammonium removal by strain F19 was evaluated using initial ammonium concentrations of 30, 50, 100, 400, 800, 1000, and 2000 mg/L as the sole nitrogen source. As shown in Figure 5A, a significant decrease in ammonium removal was observed at 30, 50, and 100 mg/L, with corresponding removal efficiencies of 96.34%, 78.24%, and 70.96%, respectively. The corresponding maximum removal rates were 0.60, 0.81, and 1.46 mg/L/h, respectively. With higher initial $NH_4^+$-N concentrations of 400, 800, 1000, and 2000 mg/L in the media, ammonium removal efficiencies were 56.38%, 34.02%, 30.18%, and 16.87%, with average removal rates of 4.05, 5.74, 6.79, 8.27, and 7.92 mg/L/h, respectively. Notably, this ammonium removal performance was comparable to that of *Bacillus methylotrophicus* L7 and *Bacillus* strain N31 [30,31], suggesting that F19 has a high ammonium tolerance and moderate removal effectively. Previously, reports speculated that a high concentration of free ammonia can be caused by increasing pH, resulting in poor removal efficiency of high-strength ammonium [38]. However, analysis of the pH effects on the ammonium removal efficiency of strain F19 showed that alkaline pH was conducive to heterotrophic nitrification with high removal efficiency. Therefore, high-strength ammonium load inhibits heterotrophic nitrification activity, resulting in low removal efficiency [38]. Accumulation of nitrate was positively correlated with the concentration of initial ammonium; as the ammonium concentration was increased from 30 to 2000 mg/L, the $NO_3^-$ concentration was significantly increased from 1.91 to 12.81 mg/L (Figure 4B). Additionally, accumulation of nitrite was almost unaffected by ammonium-N concentration. Strain F19 achieved a high removal rate of 100 mg/L ammonia (>70.96%), which is of great significance to practical applications in sewage treatment, including remediation of aquaculture water, general effluents, domestic sewerage, and other wastewater containing lower levels of ammonia.

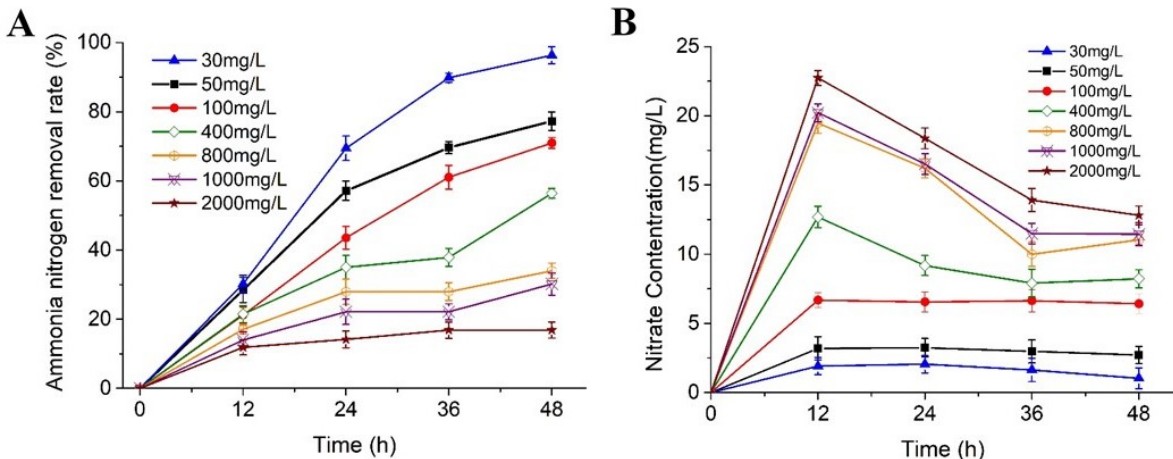

**Figure 5.** (**A**) Nitrogen removal rates and (**B**) accumulation of $NO_3^--N$ in cultures containing different concentrations of ammonia.

### 3.5. The Potential of Strain F19 for Enhancing Nitrogen Removal from Low-C/N Domestic Sewage

To verify the application of strain F19 for domestic wastewater treatment (Figure 6A), a representative low-C/N domestic wastewater with 100.00–120.00 mg/L COD, 45.00–55.00 mg/L TN, 40.00–50.00 mg/L $NH_4^+-N$, 3.00–5.00 mg/L $NO_3^--N$, and an untraceable amount of $NO_2^--N$ was used for nitrogen removal assessment. After 8 days of continuous influent, the concentrations of COD, TN, $NH_4^+-N$, and $NO_3^--N$ were significantly decreased to 32.61, 3.74, 1.05, and 2.68 mg/L, respectively, but there were undetectable amounts of $NO_2^--N$. After 20 days, the concentrations of TN and $NH_4^+-N$ in the culture system were stable (Figure 6B). However, nitrogen parameters increased rapidly due to long-term low-carbon resources, resulting in a dramatic decline in removal efficiency. When an appropriate carbon source (C/N = 2:1) was supplemented into the culture system after 20 days, concentrations of TN and $NH_4^+-N$ were further reduced to less than 4.80 and 1.20 mg/L, respectively (Figure 6B), and these levels were maintained throughout the rest of the experiments. Furthermore, there were no changes in nitrate or nitrite concentrations in the culture system. These results showed that strain F19 could efficiently remove nitrogen released from low-C/N domestic sewage without the addition of an external carbon source. These excellent performance attributes, coupled with the convenience of a single reaction chamber, make strain F19 a promising candidate for nitrogen removal from domestic wastewater.

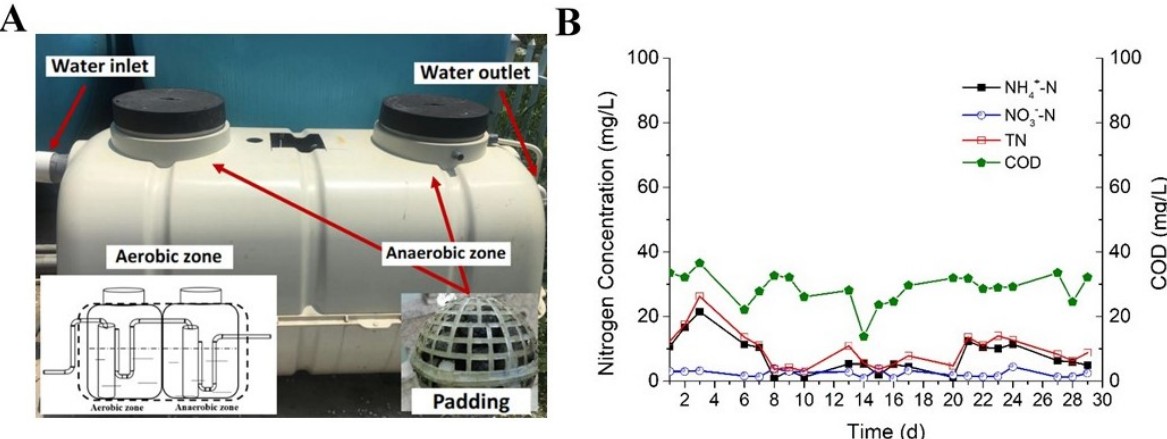

**Figure 6.** Characterization of strain F19 in a culturing and purification platform with low-C/N domestic wastewater. (**A**) Nitrogen removal purification tank. (**B**) Profile of nitrogen concentration during domestic sewage treatment with strain F19.

## 4. Conclusions

In conclusion, the novel strain F19 was isolated and identified as a species of the genus *Janthinobacterium*. F19 exhibited an impressive heterotrophic nitrifying efficiency of 96.44% during ammonium removal and achieved removal efficiencies of 92.32%, 97.46%, and 96.69% for nitrate, nitrite, and hydroxylamine, respectively. F19 could effectually remove ammonium at a low C/N ratio from domestic wastewater and high-pH wastewater. More importantly, the use of strain F19 for domestic wastewater treatment within a single chamber achieved excellent removal rates for ammonia and total nitrogen of 98.22% and 92.49%, respectively. Thus, F19 is a promising candidate for nitrogen removal from low-C/N domestic wastewater, and especially for the bioremediation of alkalized agricultural soil.

**Author Contributions:** P.J. and Z.Z. designed the experiments. Y.C. and Z.C. performed the experiments. P.J., Y.C., Z.Z., T.X., and R.Z. conceived the project, analyzed the data, and wrote the paper.

**Funding:** This work was supported by the National Natural Science Foundation of China (Grant numbers 31700078, 21276235), and the Scientific Research Foundation for Talent Program of Zhejiang Agricultural and Forestry University (W20170029).

**Conflicts of Interest:** The authors declare no competing financial interests.

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
