# Peer review of "Identification and Characterization of Janthinobacterium svalbardensis F19, a Novel Low-C/N-Tolerant Denitrifying Bacterium"

_applsci, doi:10.3390/app9091937_

Round 1
Reviewer 1 Report
At the first I would like to thank Authors for very hard work and very interesting article. Below is a few minor comments, which I believe can improve a little bit the article.
1. In my opinion the Introduction is a little bit a short. The Authors can improve the literature review. I encourage you to familiarize with two work: I) Wałęga, A.; Chmielowski, K.; Młyński D. Influence of the Hybrid Sewage Treatment Plant’s Exploitation on Its Operation Effectiveness in Rural Areas. Sustainability 2018, 10(8), 2689. II) Wałęga, A.; Chmielowski, K.; Młyński D. Nitrogen and Phosphorus Removal from Sewage in Biofilter – Activated Sludge Combined Systems. Pol. J. Environ. Stud. 2019, 28(3), 1939–1947.
2. In Introduction I would like to see what was the motivation of undertaken research.
3. In Introduction the novelty of undertaken research must be clearly emphasized. Actually there is a lack of it.
4. The figure 1 shows the Phylogenetic tree of strain F19 and close relatives based on 16S rDNA gene sequences. How the tree was constructed? Probably the cluster analysis was used. But the Authors don’t mention about it in Methodology. In revised manuscript I would like a few information about it in chapter Statistical Analysis.
5. In conclusion I would like to see how practically the results can be used.
Author Response
Response to Reviewer 1 Comments
We appreciate the reviewer #1's comments, suggestions and time on our manuscript. Those comments are all valuable and very helpful for revising and improving our paper, as well as the important guiding significance to our researches. According to these comments and suggestions, we have carefully revised and added the corresponding results in the revised manuscript which we hope meet with approval. The detailed responses to the comments are shown below.
Point 1: In my opinion the Introduction is a little bit a short. The Authors can improve the literature review. I encourage you to familiarize with two work: I) Wałęga, A.; Chmielowski, K.; Młyński D. Influence of the Hybrid Sewage Treatment Plant’s Exploitation on Its Operation Effectiveness in Rural Areas. Sustainability 2018, 10(8), 2689. II) Wałęga, A.; Chmielowski, K.; Młyński D. Nitrogen and Phosphorus Removal from Sewage in Biofilter – Activated Sludge Combined Systems. Pol. J. Environ. Stud. 2019, 28(3), 1939–1947.
Response 1: We thank the reviewer 1's valuable suggestion. As suggested, we have carefully revised the Introduction and added these references in the revised manuscript to make it richer in content.
Point 2: In Introduction I would like to see what was the motivation of undertaken research.
Response 2: Thank the reviewer 1 for this good suggestion. Accordingly, we have added more detailed description to better understand the motivation. Although many simultaneous nitrification and denitrification (SND) bacterial isolates have been reported, they pose great challenges because the removal efficiency of such bacteria is significantly inactivated in low-temperature or low-C/N wastewater environments, limiting their application. Therefore, further novel microbial resources with excellent tolerance of extreme environment (low-C/N conditions for rural domestic sewage) must be developed for efficient nitrogen removal.
Point 3: In Introduction the novelty of undertaken research must be clearly emphasized. Actually, there is a lack of it.
Response 3: Thank the reviewer 1 for this good suggestion. Accordingly, we have added more detailed description to better understand the novelty of undertaken research.
Point 4: The figure 1 shows the Phylogenetic tree of strain F19 and close relatives based on 16S rDNA gene sequences. How the tree was constructed? Probably the cluster analysis was used. But the Authors don’t mention about it in Methodology. In revised manuscript I would like a few information about it in chapter Statistical Analysis.
Response 4: Thank the reviewer 1 for the good suggestion. Accordingly, we added the mention in chapter Statistical Analysis. 16S rRNA sequence of F19 was compared to 18 representative bacterial denitrifies from the non-redundant (NR) nucleic acid sequence database (NCBI), using neighbor-joining phylogenetic analysis. One thousand bootstrap replications were performed using the MEGA software (MEGA 5.0).
Point 5: In conclusion I would like to see how practically the results can be used.
Response 5: Thank the reviewer 1 for this good suggestion. Accordingly, we have carefully revised the manuscript.

Reviewer 2 Report
Comments for authors:
This study is mainly about isolating useful strains of bacteria concerning wastewater treatment plant. Based on the result present in the article, the isolated Janthinobacterium svalbardensis F19 can be widely use in treatment plant to improve the treatment of wastewater with low C/N ratio.
In the following I put some comment that is worth to consider:
1- The mechanism of nitrogen removal with this strain was not presented. I am wondering what is this strains, as the author mention it can do nitrification and denitrification with source of carbon. I f the strain are like the one used in anammox the author should provide couple of reference on that.
2- As I mentioned, the author should provide a figure explaining the mechanism of nitrogen removal by the isolated strains
3- Different sources of carbon were used in this study, what is the reason behind the effectiveness of sodium acetate. If bicarbonate as the source of carbon cause approximately 50% of ammonia removal, can you stated that 50% of strains are autrotrophic.
4- In material and method, you should indicated that how you confirm the aerobic or anaerobic condition. Did you measure dissolved oxygen concentration.
5- In part 2.3., some statement should be present in result part, especially from line 90 to line 95.
6- There are some misunderstanding in the concentration of nitrogen in line 111. It is 40 mg N-NH4 or 40 mg NH4 or 40 mg NH4Cl/L
7- It is very important to filter the wastewater before putting it into the purified strain. Did you do that?
8- How do you provide air in your culture. Did you measure DO concentration?
9- Isolation procedure should be written in more details.
10- Clarification of heterotrophic/autotrophic bacteria in bacterial community should be provided to have a better view why in higher C/N ratio, you have more nitrification.
11- The author should organize the result based on the kinetic study. presenting rate of nitrogen removal by mgN/L/g bacteria/h is better to give us the idea of what is the concentration of bacteria.
12- It seem that in temperature higher than 25 degree the efficiency should be declined as the Dissolve Oxygen was diminishing rapidly.
Author Response
Response to Reviewer 2 Comments
We appreciate the editor and reviewers’ comments, suggestions and time on our manuscript. Those comments are all valuable and very helpful for revising and improving our paper, as well as the important guiding significance to our researches. According to these comments and suggestions, we have carefully revised and added the corresponding results in the revised manuscript which we hope meet with approval. The detailed responses to the comments are shown below.
Point 1: The mechanism of nitrogen removal with this strain was not presented. I am wondering what is this strain, as the author mention it can do nitrification and denitrification with source of carbon. If the strains are like the one used in anammox the author should provide couple of reference on that.
Response 1: Thank the reviewer 2 for this good suggestion. Recently, a group of bacteria capable of simultaneous nitrification and denitrification (SND) were reported (Yao et al., Bioresource Technology 139 (2013) 80–86; He et al., Bioresource Technology 200 (2016) 493–499; Huang et al., Journal of Bioscience and Bioengineering, (2017)…. ). These heterotrophic bacteria display higher growth rates than autotrophs and can use organic substrates as sources of carbon and energy to convert ammonium into nitrogenous gas under aerobic conditions. Researches mainly focused on separation, identification and application of single functional strain. However, there is a great challenge that the metabolic pathways and mechanisms of simultaneous nitrification and denitrification have not yet been unclarified and systematically investigated. We also carried out research on nitrogen metabolism of SND in our recent study (Jin et al., Journal of Hazardous Materials, 371 (2019), 295–303). Therefore, we have revised the manuscript and added relevant literatures.
Point 2: As I mentioned, the author should provide a figure explaining the mechanism of nitrogen removal by the isolated strains
Response 2: Thank the reviewer 2 for this good suggestion. Accordingly, we have added a figure to predicted nitrogen utilization pathway in Janthinobacterium svalbardensis F19. (Fig. 4E).
Point 3: Different sources of carbon were used in this study, what is the reason behind the effectiveness of sodium acetate. If bicarbonate as the source of carbon cause approximately 50% of ammonia removal, can you stated that 50% of strains are autotrophic.
Response 3: We thank the reviewer 2’s valuable suggestion. Different carbon sources have obvious differences in the utilization rate of microbial metabolism, which leads to a great deviation in the removal efficiency of ammonia nitrogen by bacterial strains. (such as Yang et al., Bioresource Technology, 256 (2018), 366-373; Duan et al., Bioresource Technology, 179(2015), 421-428; Liu et al., Chinese Journal of Chemical Engineering, 23(2015), 827-834; ……).
Point 4: In material and method, you should indicate that how you confirm the aerobic or anaerobic condition. Did you measure dissolved oxygen concentration.
Response 4: Thank the reviewer 2, we have revised this operation in the part 2.5.
Point 5: In part 2.3., some statement should be present in result part, especially from line 90 to line 95.
Response 5: Thank the reviewer 2, we have carefully revised this statement according to the suggestion.
Point 6: There are some misunderstanding in the concentration of nitrogen in line 111. It is 40 mg N-NH4 or 40 mg NH4 or 40 mg NH4Cl/L
Response 6: Thank the reviewer 2 for the careful observation. In the revised manuscript, we have modified it.
Point 7: It is very important to filter the wastewater before putting it into the purified strain. Did you do that?
Response 7: We thank the reviewer 2’s good suggestion. At the front end, the wastewater got through the septic tank treatment and simple physical filtration, and then directly entered the system continuously and stably. Therefore, our system was an open system, in which we conducted efficient and stable nitrogen removal by pre-culture and colonization of the dominant strain population in the system. This pattern has been reported in our recent study (Jin et al., Bioresource Technology, 281 (2019), 392–400).
Point 8: How do you provide air in your culture. Did you measure DO concentration?
Response 8: Thank the reviewer 2. Accordingly, we have added the measurement method of Dissolved oxygen in Analytical methods.
Point 9: Isolation procedure should be written in more details.
Response 9: Thank the reviewer 2 for this good suggestion. Accordingly, we have added more detailed description to better understand the operation.
Point 10: Clarification of heterotrophic/autotrophic bacteria in bacterial community should be provided to have a better view why in higher C/N ratio, you have more nitrification.
Response 10: Thank the reviewer 2 for this good suggestion. In fact, the bacterium in this experiment is a simultaneously heterotrophic nitrifying–aerobic denitrifying. This kind of bacteria can be denitrified under both aerobic and anaerobic conditions. Of course, bacterial 16s rDNA sequencing and evolutionary tree analysis were used to determine its classification in manuscript.
Point 11: The author should organize the result based on the kinetic study. Presenting rate of nitrogen removal by mgN/L/g bacteria/h is better to give us the idea of what is the concentration of bacteria.
Response 11: Thank the reviewer 2 for this good suggestion. As suggestion, we also modified the representation of some removal efficiency in manuscript. In the flask shaking experiment, the biomass and metabolic activity of the bacteria were continuously increased, while the removal ability decreased with the consumption of carbon source and other nutrients in the stable period, which was difficult to be well described the nitrogen removal by mg N/L/g bacteria/h. In our system, it is a continuous process of inlet and outlet water, and it is difficult to determine the weight of the bacteria in the open system. Therefore, we also used simple and common description with referring other literatures.
Point 12: It seem that in temperature higher than 25 degree the efficiency should be declined as the Dissolve Oxygen was diminishing rapidly.
Response 12: Thank the reviewer 2 for this good suggestion. On the one hand, we found that the change of DO caused by the temperature difference in the shaking bottle had little effect, and it would not have a great impact on the removal efficiency. On the other hand, temperature, the removal efficiency of Higher than 25 °C decreased slightly, is probably the most suitable temperature for the bacteria to nitrogen removal is 25 °C - 30 °C. The depletion of the carbon source may result in the maximum removal efficiency remaining at this level.

Round 2
Reviewer 1 Report
The manuscript has been significantly improved following the recommendations of the Reviewers; all my concerns have been addressed and convincingly justified. In my opinion the paper can be accepted for publication in the present form.
Reviewer 2 Report
The author correct all the mentioned comments.